# Influence of Dietary Supplementation of *Salix alba* Bark on Performance, Oxidative Stress Parameters in Liver and Gut Microflora of Broilers

**DOI:** 10.3390/ani10060958

**Published:** 2020-05-31

**Authors:** Tatiana Dumitra Panaite, Mihaela Saracila, Camelia Puia Papuc, Corina Nicoleta Predescu, Cristina Soica

**Affiliations:** 1National Research-Development Institute for Animal Biology and Nutrition (IBNA), 1, Calea Bucuresti, Balotesti, 077015 Ilfov, Romania; tatiana.panaite@ibna.ro (T.D.P.); cristina05101988@yahoo.com (C.S.); 2Faculty of Veterinary Medicine, University of Agronomic Sciences and Veterinary Medicine, 105 Splaiul Independentei, 050097 Bucharest, Romania; cami_papuc@yahoo.com (C.P.P.); durduncorina@yahoo.com (C.N.P.); 3Academy of Romanian Scientists (AOSR), 54 Splaiul Independentei, 050094 Bucharest, Romania

**Keywords:** broiler, *Salix alba* bark, liver, enzyme, antioxidant, gut microflora

## Abstract

**Simple Summary:**

Nowadays, due to genetic evolution, poultry is more susceptible to stressful factors (nutritional, environmental, etc.), which causes imbalances in the redox status, namely oxidative stress. Among the consequences of oxidative stress in poultry are impaired performances, gut microflora imbalance, as well as the deterioration of the overall health status. Consumer pressure about the quality and safety of animal products has prompted the need for use, among many dietary factors, of natural antioxidants. *Salix alba* bark represents a source of bioactive compounds, including polyphenols and flavonoids, with an important role in poultry nutrition. This study aims to describe the effects of dietary *Salix alba* bark powder on the performance of oxidative stress parameters in liver and gut microflora of broilers. As conclusion, the paper proposes that dietary 0.05% *Salix alba* bark powder could be an effective solution to impede the oxidative stress in broiler liver and to improve gut microflora. The paper manifests noteworthy concern in poultry nutrition, whereas many diseases have been related to oxidative stress.

**Abstract:**

The paper aimed to analyse the effect of dietary *Salix alba L.* bark powder (SAB) on broiler performance in terms of oxidative stress parameters in liver and gut microflora. One hundred and eighty Cobb 500 broiler chicks (14 days) were allotted to three homogeneous treatments (SAB 0%; SAB 0.025%; SAB 0.05%). The broilers were housed in an environmentally controlled space (10 replicates, six broilers/replicate). Compared to dietary control treatment (SAB 0%), the other treatments included 0.025% SAB (SAB 0.025%) and 0.05% SAB (SAB 0.05%). The results showed that SAB powder used in broiler diet had a high total phenolic content. Regarding the performance results, significant differences between experimental and control treatments were recorded only for average daily feed intake (35–42 days). The broilers fed with SAB powder had a significantly lower hepatic level of malondialdehide and glutathione, a higher total antioxidant capacity than those fed control treatment, and demonstrated a positive effect on the development of non-pathogenic bacteria (*lactobacilli*) but a decrease in the population of pathogenic ones (*E. coli*, *staphylococci*). Our findings suggested that dietary 0.05% SAB powder could be an effective solution to impede the oxidative stress in broiler liver and to improve gut microflora.

## 1. Introduction

Nowadays, commercial poultry production is subject to many stressors which consequently impair the productive parameters of growing chicks. There are evidences showing that most of the stresses (nutritional, physiological/pathological, or environmental factors) found in the poultry sector lead to oxidative stress [1]. Oxidative stress starts from the disequilibrium between reactive oxygen species (ROS) production and antioxidant defense, consequently leading to cellular and tissue injury as lipid peroxidation, protein alteration, etc. [2]. Thus, research is needed for solutions to prevent or alleviate the oxidative stress. The current trend is using plants with antioxidant properties, which were exploited by traditional medicine as treatments or as ways of preventing other diseases. This trend is particularly important since the administration of antibiotics as growth promoters has been forbidden. In recent years, many plants containing phenolic compounds have gained considerable interest for reducing oxidative stress both in vitro and in vivo studies [1,3,4]. 

White willow bark (*Salix alba L*.) has antipyretic and analgesic properties, due mainly to salicin content [5]. The parts of willow species which can be used are mainly represented by bark and leaf. Besides the salicin content, willow also represents a source of bioactive compounds, including polyphenols and flavonoids, with significant implications in fulfilling the curative roles of willow [6]. 

So far, there are many in vivo and in vitro studies that have proved the antioxidant properties of willow extracts [7,8,9]. A good antibacterial activity of *Salix alba* was reported [10], positively correlated with its antioxidant potential [11]. In a study on humans, it was concluded that willow bark is able to increase the activity of antioxidant enzymes and consequently counteract oxidative reactions [12]. Some authors noticed that using 1% dietary hydroglyceroalcoholic willow bark extract [7] and 0.05% willow bark extract powder [4] improved the biochemical blood parameters and liver oxidative status of heat-stressed broilers and decreased the number of pathogens in the caecum.

The purpose of this paper was to describe the effects of dietary extract of *Salix alba* bark powder on the productive parameters of oxidative stress markers in the liver and gut microflora of broilers.

## 2. Materials and Methods

The experimental trial followed the regulations of Directive 2010/63/EU [13] concerning animal’s protection used for scientific purposes and was certified by the Ethical Commission (no. 52/30.07.2014) of National Research Development Institute for Animal Biology and Nutrition (IBNA-Balotesti, Balotesti, Romania).

### 2.1. Experimental Design

An experiment that lasted for 4 weeks was carried out on 180 1-day-old Cobb 500 broiler chicks ($\tilde{x}$ = 40.40 ± 2.3 g) that were provided by a trading hatchery. They were sheltered in an environmentally verified space (temperature, humidity, ventilation, and light program) with a capacity of 16 broilers/m^2^. As permanent broiler litter, wood shavings with 10–12 cm thickness were used. For the third day, the room temperature was maintained at 33 °C and then reduced gradually. Through 14 days of age, broilers were fed with a commercial diet based on corn (56%) and soybean meal (32%) with 23% CP and 3039.79 kcal/kg ME. At 14 days of age, depending on the body weight, the chicks were allotted to three homogeneous treatments (SAB 0%, SAB 0.025% and SAB 0.05%) in terms of body weight: 360.21 ± 8.69 g (SAB 0%), 360.30 ± 9.73 g (SAB 0.025%), and 360.67 ± 7.37 g (SAB 0.05%). Every group was run in ten replicates (6 chicks/replicate). Feed and water were administered ad libitum. Diets structure was considered to follow the requirements stipulated in NRC [14] By comparison with the commercial diet (SAB 0%), the experimental treatments included 0.025% of *Salix alba* bark (SAB) powder for SAB 0.025% treatment and 0.05% SAB powder for SAB 0.05% treatment (Table 1). The SAB powder containing 25–98% salicin was purchased from a commercial company in China (Changsha Vigorous-Tech Co., Ltd., Changsha, China). Over the experimental trial, the chicks were not subjected to any medical program or medical treatment. 

### 2.2. Performance

During the experimental trial (14–42 days), bodyweight (BW, g) and average daily feed intake (ADFI, g feed/broiler/day) were monitored while average daily weight gain (ADWG, g/broiler/day), and feed conversion ratio (FCR, g feed/g gain) were calculated. Individual bodyweight was recorded weekly. The experiment protocol stipulated that mortality should be recorded daily throughout the experiment.

### 2.3. Collection of Blood, Liver Samples and Intestinal and Caecal Content

At 42 days of age, and before slaughter, one bird from each replicate had their blood aseptically collected from the brachial vein into vacutainers including anticoagulant (lithium heparin). These samples were used to determine the serum biochemical parameters. After blood sampling, one bird from each replication, with bodyweight within ±10 g standard deviation of the mean treatment weight, was euthanized by dislocation of the cervical spine. After bleeding, the excision of internal organs was performed, including the gut (from the oesophagus to the cloaca). Liver samples were collected according to [15] and then packed under vacuum conditions and preserved at −80 °C until the analysis of biochemical markers of oxidative stress. The contents collected from the small intestine (duodenum, jejunum and ileum) and caecum were aseptically stored at −20 °C until performing the microbiological analyses (*E. coli*, staphylococci, *lactobacilli*, *Salmonella spp*).

### 2.4. Total Phenolic Content (TPC) in SAB Powder

The ethanolic extract was obtained from 1 g of ground plant material and 10 mL of solvent (60% *v/v* ethanol, 40% distilled water). The TPC of SAB powder was determined according to a modified Folin-Ciocalteu method [16]. The absorbance value was recorded at 725 nm wavelength against blank sample. The TPC were indicated as gallic acid equivalents/g dry weight (GAE/g dw).

### 2.5. Analyses of Serum Biochemical Parameters

Blood samples were prepared by centrifugation at 775× *g* for 25 min at 4 °C. The supernatant obtained was employed to analyse the following serum markers: glucose, cholesterol, triglyceride, total bilirubin, total protein, alanine aminotransferase, and aspartate aminotransferase. The biochemical parameters were analysed using an automatic BS-130 Chemistry analyzer (Bio-Medical Electronics Co., Ltd., Shenzhen, China).

### 2.6. Liver Oxidative Stress Evaluation

The sample preparation was done according to [17]. One gram of liver tissue was homogenized in 10 mL of cold 66 mM potassium phosphate buffer, pH 7.2 with 1 mM EDTA. After centrifugation (10,000× *g*, 15 min, 4 °C), the supernatants were taken and used for analysing the lipid peroxidation, estimated by the spectrophotometric method of thiobarbituric acid reactive substances (TBARS), protein oxidation (PC), glutathione (GSH), total antioxidant capacity (TAC), and superoxide dismutase (SOD).

#### 2.6.1. Lipid Peroxidation Evaluation (TBARS Method)

Lipid peroxidation was evaluated using a modified assay of [18]. One hundred microliters of liver extract were homogenized using 400 μL of Tris-HCl (20 mM; pH 7.4) and 500 μL of 20% trichloroacetic acid and incubated for 10 min at room temperature. After the addition of 1 mL thiobarbituric acid (TBA), the samples were incubated at 95 °C, 45 min. then, were cooled down in ice-water bath (5 min.). They were centrifuged at 10,000× *g*, 10 min. and the absorbance was recorded at 532 nm wavelength. The thiobarbituric acid reactive substances (TBARS) concentration was calculated using the molar extinction coefficient of malondialdehyde and expressed as nmol/g tissue.

#### 2.6.2. Protein Oxidation (PO)

PO was estimated by the spectrophotometric determination of protein carbonyl (PC). PC was determined following the method of [19]. Fifty microliters of 20% TCA were homogenized with 50 μL of liver extract and kept at incubation in ice-water bath, 15 min. Consequently, the samples were submitted to centrifugation (15,000× *g*, 10 min, 4 °C).

Then, after removing the supernatant, 500 μL of 2,4-dinitrophenylhydrazine (14 mM, prepared in 2.5 N HCl) was added and incubated for 60 min at room temperature, in the dark. In parallel, a blank was prepared, adding 500 μL of 2.5 N HCl. After centrifugation, the supernatant was removed and 1 mL of TCA (10%) was added. The pallet was washed twice with 1 mL of ethanol-ethyl acetate (1:1, *v/v*) each time. Then, 1 mL of urea (5 M, pH 2.3) was added to the washed pallet and the mixture was incubated for 15 min at 37 °C. The absorbance was read at 375 nm. PC concentration was calculated using molar extinction coefficient. The results were expressed as nmol/g tissue.

#### 2.6.3. Glutathione (GSH) Estimation

GSH was performed according to the method described by [20]. Twenty microliters of liver extract homogenized with 5% trichloroacetic acid were added to 650 μL of 66 mM potassium phosphate buffer (pH 8.0) and 330 μL of 1mM 5, 5 dithiobis-2-nitrobenzoate (DTNB). Consequently, the samples were stored in a dark place for 45 min. and after this were incubated. The absorbance values of samples were recorded at 412 nm wavelength. Glutathione values were estimated using a calibration curve, performed utilizing commercial standards. The results were indicated as μmol/g tissue.

#### 2.6.4. Total Antioxidant Capacity (TAC) Estimation

The estimation of TAC was performed using the protocol described by [21]. Twenty microliters of liver extract were homogenized with 480 μL of 22 mM potassium phosphate buffer (pH 7.4) and 500 μL of 0.1 mM 2,2-diphenyl-1-picrylhydrazyl (DPPH) free radical. After this, samples were kept at incubation in a dark room for 30 min and centrifuged for 3 min. at 10,000× *g*. The absorbance of samples was recorded at 520 nm wavelength. The DPPH· scavenging capacity was calculated using the following formula:Radical scavenging activity (%) = (A_blank_ − A_sample_)/A_blank_ × 100(1)
where A = Absorbance.

#### 2.6.5. Superoxide Dismutase Activity (SOD) Estimation

An SOD Assay Kit (Sigma, Buchs, Switzerland) was used for determining the superoxide dismutase (SOD). The results were calculated following the producer’s details. They were indicated as U/g tissue.

### 2.7. Microbiological Analyses

The microbiological analyses of *E. coli*, *staphylococci*, *lactobacilli* and *Salmonella* spp. were determined as [22] described. The Scan 300 colony counter (Interscience, Paris, France) was used to establish the colony forming units of *E. coli*, *staphylococci*, and *lactobacilli*. The results were expressed as log base 10 colony-forming units (CFU) per gram of intestinal/caecal contents.

### 2.8. Statistical Analysis

A complete randomized model was performed to analyse the results for productive parameters, biochemical parameters, liver oxidative status, and gut microflora. The effects of treatments were tested by analysis of variance (ANOVA) using STATVIEW for Windows (SAS, version 6.0). When the F-test results were significant, differences between means were declared as *p* < 0.05.

## 3. Results

### 3.1. Total Phenolic Content (TPC) in Salix alba Bark

Salix alba powder used in this study has a high level of TPC, 98.5 ± 2.2 mg GAE/g, respectively.

### 3.2. The Effect of Salix alba Bark on Broiler Performance

Table 2 shows the productive parameters of broilers fed treatment with and without SAB powder addition.

At both 35 and at 42 days, broilers fed SAB 0.05% recorded a higher (*p* < 0.05) BW compared with the broilers fed SAB 0.025% treatment, but comparable with broilers fed SAB 0%. Only in the last week of the trial (35–42 days) the ADFI was significantly higher (*p* < 0.05) in SAB 0.05% treatment compared with both SAB 0% and SAB 0.025%. Throughout the entire experimental trial, no significant differences were recorded between treatments (Table 2). No significant differences were noticed for SAB 0.025% and SAB 0.05% compared to SAB 0% treatment regarding ADWG throughout entire experimental period (Table 2). The values recorded for SAB 0.05% treatment were significantly increased compared to SAB 0.025%. There was no effect (*p* > 0.05) of SAB powder on FCR (14–42 days). Even though, in the last week of the trial, the broilers from SAB 0.05% group had a higher ADFI (*p* < 0.05) than the other groups, the FCR was similar to SAB 0% and lower than SAB 0.025% (*p* > 0.05).

Throughout the entire experimental period, no mortalities were recorded between the three treatments.

### 3.3. The Effect of Salix alba Bark on Serum Biochemical Measurements

Regarding serum biochemical measurements (Table 3), the administration of SAB powder (SAB 0.025 and 0.05%) decreased (*p* < 0.05) the glucose level compared with no SAB addition diet (SAB 0%).

The serum cholesterol and triglyceride concentrations did not recorded differences (*p* > 0.05) between the three treatments. Both ALT and AST levels were reduced (*p* < 0.05) in the serum of broilers fed treatment with higher dose of SAB powder (0.05%) than in that of serum broilers fed no SAB addition diet (SAB 0%) or with 0.025% SAB powder addition.

### 3.4. The Effects of Salix alba Bark on Oxidative Stress Biomarkers in Liver

The oxidative stress biomarkers in the liver tissue of chicken fed commercial diet and those fed SAB powder treatment are shown in Table 4.

Lipid peroxidation indicator (TBARS values) was lower (*p* ˂ 0.05) in the liver tissue from chickens fed SAB powder treatments (SAB 0.025% and SAB 0.05%). Both doses of inclusion of the SAB powder (0.025%, 0.050%) inhibited protein oxidation (*p* < 0.05), but the lowest concentration of PC was found in the liver tissue of the chicken fed SAB 0.05% treatment. Adding SAB powder (0.05%) in feed significantly increased TAC and GSH concentrations in chicken liver tissue (Table 4). The addition of SAB powder in broiler feed did not significantly affect the SOD activity in liver tissues.

### 3.5. The Effects of Salix alba Bark on Intestinal and Caecal Bacterial Populations

The microbiological analyses on caecum content (Table 5) revealed that birds fed SAB 0.05% treatment had a lower colony forming number of *E. coli* and *staphylococci* than in the intestinal and caecal content of those from the control group.

There were no differences (*p* > 0.05) between SAB 0.025% and SAB 0% treatments on pathogenic bacteria tested (*E. coli*, *staphylococci*). Birds fed diets supplemented with SAB powder had higher intestinal and caecal *lactobacilli* populations compared with those fed SAB 0% treatment (*p* < 0.05). Different levels of SAB powder significantly increased intestinal and caecal *lactobacilli* populations. *Salmonella* spp. were absent in all samples.

## 4. Discussion

Data on total phenolic content of *Salix alba* powder used in this study revealed a high level of TPC. There are many studies showing that *Salix alba* bark is a natural source of phenolic compounds with antioxidant activity [8,9,11]. Compared to data reported by [6] on total phenolic content of ethanolic extracts of *S. alba* bark, our results are 2.4 times higher. Also, ethanolic extract of *S. alba* and *S. purpurea* bark are reported to contain important concentration of phenolic compounds [23]. Among phenolic compounds presented in *Salix* bark species, the flavonoids are found in the highest quantities [23]. Regarding the performance results (Table 2), significant differences between experimental treatments (SAB 0.025%, SAB 0.05%) and SAB 0% were recorded only for ADFI (35–42 days). However, the level of 0.05% SAB powder had higher efficacy compared to 0.025% SAB powder at any stage of the bird’s life. The higher BWG was recorded in broilers fed SAB 0.05% treatment. Therefore, others [24] found that BWG of broilers decreased when there were added 400 mg/kg of *Salix alba*. Under heat stress, diets supplementation with 0.025% and 0.05% hydroalcoholic extract of willow bark powder did not have a significant (*p* > 0.05) influence on broiler performance (14–42 days) [7]. A significantly improvement in BWG (8–43 days) was reported by [25] when broilers were fed 1000 mg/L willow bark extract in the drinking water. In the present work, the SAB 0.05% treatment significantly improved (*p* < 0.05) the broilers’ performance, compared to SAB 0.025% treatment. This idea was supported also by other researchers. For example, in a study conducted on rabbits [9] it was reported that a higher dose of *Salix alba* extract (120 mg/kg diet) was more effective than the lower dose (60 mg/kg diet). Similar results were found on heat-stressed broilers [4] when willow bark extract was used (50 g/100 kg diet). The explanation for this result may be that SAB acts in a dose-dependent mode, an idea highlighted by other researchers [9,26]. The dose of inclusion for phenolic compounds can vary considerably, but the nutritional impact and biological effects depend on their bioavailability in the digestive tract [27]. In this regard there are plants that showed beneficial effects at high doses of inclusion and others, at low doses, depending on physicochemical characteristics of the phenolic compound [27].

The serum biochemical parameters demonstrated the advantages of broiler feeding with SAB powder. The serum glucose level decreased (*p* < 0.05) in the treatments fed including SAB powder. A reduction in serum glucose may be due to the mechanism of polyphenols in regulating glucose homeostasis and in improving insulin sensitivity, an idea supported by in vitro and in vivo medical studies of many researchers [28,29,30]. These results may account for assessment of SAB powder’ contribution in maintaining the overall health of broiler. Diet supplementation with 1% willow bark extract had an hypocholesterolemiant and hypoglycaemic effect in the heat-stressed broilers [7]. A study performed on normoglycaemic mice showed the property of *Salix tetrasperma* extract to reduce the blood glucose concentration than the control diet [31]. Both, ALT and AST serum levels indicate the health status of liver [32]. Hepatic enzyme activity of broilers fed higher dose of SAB powder (SAB 0.05%) was improved by decreasing (*p* < 0.05) the serum levels of ALT and AST. These findings are in compliance with those stated by [7]. They found that ALT and AST levels in the serum of heat-stressed broilers fed 1% willow bark extract diet were significantly lower compared to those fed conventional diet. An explanation in this regard might be that polyphenols from SAB contribute to the protective effect on hepatic function. Generally, this effect of polyphenols has also been extensively demonstrated, majorly via antioxidative stress and anti-inflammation [33]. Also, others [9] reported that *Salix alba* extract could act as free radical scavenger, the effect being directly proportional with the dose of inclusion.

The results on oxidative stress biomarkers highlighted that dietary supplementation with polyphenolic powder of SAB ameliorated the liver oxidative status. This finding is important as long as in animal production there are many diseases related to oxidative stress [34]. Lipids are susceptible targets of ROS due to the large number of reactive double bonds. Thiobarbituric acid reactive substances (TBARS) are carbonyl compounds generated in vivo via peroxidation of polyunsaturated fatty acids. Our results show that the dietary SAB powder is able to reduce the lipid peroxidation in chickens. In the literature, there were recorded decreases in liver malonaldehyde content for broilers fed dietary hydroalcoholic extract of willow bark powder (0.025% and 0.05%, respectively) compared to those fed the conventional diet under heat stress conditions [4]. This property of *Salix* bark is probably attributed to its high antioxidant content, scavenging free radicals, and consequently diminishing the oxidative stress.

Protein carbonyl (PC) groups (e.g., aldehides, ketones) are generated as consequence of amino acids oxidation or through oxidative cleavage of proteins [35]. In our study, PC levels were decreased significantly in SAB 0.025% and SAB 0.05% treatments, compared to SAB 0%. Moreover, the decrease was dose-dependent, suggesting that it was stronger in the group fed high level of SAB powder. Since protein is the most important nutrient for animal growth, its oxidation damage protein structure and function, consequently impairing performance [36]. Intracellular glutathione (GSH) as a biomarker of oxidative stress has a meaningful purpose to fight against the cellular damage induced by ROS, occurring within a cell during chemical reactions. When cells are exposed to reactive species, GSH are the first consumed antioxidants [37]. Notably is that phenolics have many important properties, as antioxidant [38] and anti-inflammatory properties [39]. Data from the present study revealed only the high dose of SAB powder expresses a significantly antioxidative protection in broiler liver. Other studies also demonstrated that willow bark extract impaired oxidative stress by increasing GSH in animal [40,41,42] and human [12] models. SAB powder stimulated the biosynthesis and secretion of antioxidant enzyme, scavenging the free radicals as reflected by increasing the activity of GSH.

The ingestion of natural antioxidants has been associated with an increase in the cell’s antioxidant activity [43]. The total antioxidant capacity (TAC) shows the body or tissue antioxidant properties. The present results showed that the dietary SAB powder improved the antioxidant status in broiler. In particular, TAC was increased as a consequence of the level of augmentation of the dietary SAB powder (i.e., 0.05%). The abovementioned increase in TAC for experimental groups could be attributed to the stimulating effect of SAB powder on antioxidant GSH activity in liver.

The antioxidant enzyme, superoxide dismutase (SOD) had a crucial implication in annihilating H_2_O_2_ and O_2_^−^ oxidation injury at the cellular level [44]. The effects of SAB powder on SOD activity showed that the nutritional supplement used in our experiment did not affect SOD activity in the liver. Others [40,45] showed that willow bark extract overcomes oxidative stress by improving the activity of antioxidant enzymes and acting as a radical scavenger. Also, dietary *S. alba* extract and acetylsalicylic acid enhanced the SOD activity and GSH levels, while malondialdehide concentrations were decreased in the tissues of hypercholesterolaemic rabbits [9]. Notably, the effectiveness of phenolic compounds in the increase of antioxidant enzymes activity is dependent on the chemical structure, dose, and bioavailability [1].

The present study revealed the efficacy of SAB 0.05% treatment in reducing the pathogens like *E. coli* and *staphylococci* and multiplied the *lactobacilli* both in the caecum and intestinal segment. The lower dose of SAB (0.025%) had effect (*p* < 0.05) only on *lactobacilli*, increasing their intestinal and caecal number. It seems like *lactobacilli* are sensitive to SAB action even at a low dose of inclusion in broiler diet. There were many studies proving the antibacterial effect of *Salix alba* both in vitro and in vivo [4,7,10,11]. Among in vivo studies, only a few of them were conducted on broilers. For example, some authors [7] showed that, even under heat stress, the dietary hydroglyceroalcoholic willow bark extract (1%) significantly decreased *E. coli* and *staphylococci* populations in broiler caeca at 42 days.

The effect of SAB powder in this study might be attributed to its polyphenolic content and antioxidant activity. This can be explained by the fact that *lactobacilli* are more susceptible to polyphenol action and possess the capacity to decompose phenolics, delivering the energy to cells, therefore supporting bacteria growth [46]. Phenolic compounds are recognized for their direct effectiveness against pathogenic bacteria or impairment of the bond of pathogens in the intestine [47]. The mechanism by which polyphenols act against the pathogenic bacteria is not fully clear. It might be by lowering the pH of the intestinal environment, due to the capacity of these compounds to chelate iron, vital for the survival of almost all bacteria [48], etc. Thus, the bioactive compounds of SAB powder can serve to overcome the multiplication of pathogenic bacteria and stimulate the development of the non-pathogenic ones (e.g., *lactobacilli*) only at a higher dose (0.05%).

## 5. Conclusions

The paper highlighted that the dietary polyphenolic powder of *Salix alba* bark was able to improve the oxidative status of broilers by promoting the protective activity of antioxidant enzymes against oxidation. The improvement noticed on gut composition could be attributed to the phenolic compounds of SAB, which could be useful to preserve the equilibrium between bacteria.

## Figures and Tables

**Table 1 animals-10-00958-t001:** Diet formulation.

Specification	Grower Stage (14–35 days)			Finisher Stage (36–42 days)		
	SAB 0%	SAB 0.025%	SAB 0.05%	SAB 0%	SAB 0.025%	SAB 0.05%
	%			%		
Corn	62.00	61.97	61.95	60.50	60.47	60.45
Soybean meal (46% CP *)	26.58	26.58	26.58	25.46	25.46	25.46
Gluten	4.00	4.00	4.00	6.00	6.00	6.00
Oil	2.50	2.50	2.50	3.75	3.75	3.75
*Salix alba* bark powder	0	0.025	0.05	0	0.025	0.05
Calcium carbonate	1.40	1.40	1.40	1.33	1.33	1.33
Monocalcium phosphate	1.36	1.36	1.36	1.13	1.13	1.13
Salt	0.37	0.37	0.37	0.33	0.33	0.33
DL-Methionine	0.26	0.26	0.26	0.25	0.25	0.25
Lysine HCl	0.48	0.48	0.48	0.20	0.20	0.20
Choline	0.05	0.05	0.05	0.05	0.05	0.05
Premix **	1	1	1	1	1	1
Total	100	100	100	100	100	100
Calculated Metabolisable energy, kcal/kg	3140.03	3140.03	3140.03	3250.00	3250.00	3250.00
*Chemical composition—calculated*						
Dry matter, %	86.48	86.48	86.48	86.49	86.49	86.49
Crude protein, %	20.5	20.5	20.5	19.50	19.50	19.50
Ether extract, %	4.46	4.46	4.46	5.66	5.66	5.66
Crude fibre, %	3.54	3.54	3.54	3.56	3.56	3.56
Calcium, %	0.84	0.84	0.84	0.78	0.78	0.78
Phosphorus, %	0.75	0.75	0.75	0.74	0.74	0.74
Available phosphorus, %	0.42	0.42	0.42	0.39	0.39	0.39
Lysine, %	1.35	1.35	1.35	1.15	1.15	1.15
Methionine, %	0.58	0.58	0.58	0.59	0.59	0.59
Meth + Cys, %	0.92	0.92	0.92	0.95	0.95	0.95
Threonine, %	0.74	0.74	0.74	0.77	0.77	0.77
Triptophan, %	0.21	0.21	0.21	0.22	0.22	0.22

* %CP of DM. ** 1 kg premix contains 1,100,000 IU/kg vit. A; 200,000 IU/kg vit. D3; 2700 IU/kg vit. E; 300 mg/kg vit. K; 200 mg/kg Vit. B1; 400 mg/kg vit. B2; 1485 mg/kg pantothenic acid; 2700 mg/kg nicotinic acid; 300 mg/kg vit. B6; 4 mg/kg Vit. B7; 100 mg/kg vit. B9; 1.8 mg/kg vit. B12; 2000 mg/kg vit. C; 8000 mg/kg manganese; 8000 mg/kg iron; 500 mg/kg copper; 6000 mg/kg zinc; 37 mg/kg cobalt; 152 mg/kg iodine; 18 mg/kg selenium. CP = Crude protein; DM = dry matter.

**Table 2 animals-10-00958-t002:** Effect of dietary SAB powder on broiler productive parameters.

Parameter	Age	SAB 0%	SAB 0.025%	SAB 0.05%	SEM	*p*-Value
Body weight (g)	14 days	360.21	360.30	360.67	4.950	0.9992
	35 days	2187.25 ^ab^	2121.25 ^a^	2280.88 ^b^	22.473	0.0267
	42 days	2822.21 ^ab^	2714.85 ^a^	2968.53 ^b^	37.350	0.0195
Average daily feed intake (g feed/broiler/day)	14–35 days	128.85	127.94	134.79	3.522	0.6965
	35–42 days	162.06 ^a^	153.14 ^b^	174.68 ^c^	2.344	0.0356
	14–42 days	136.15	133.50	144.10	3.066	0.3444
Average daily weight gain (g/broiler/day)	14–35 days	87.00 ^ab^	83.86 ^a^	91.43 ^b^	1.101	0.0335
	35–42 days	90.71 ^ab^	84.8 ^a^	98.24 ^b^	2.257	˂0.0001
	14–42 days	87.93 ^ab^	84.09 ^a^	93.14 ^b^	1.357	0.0301
Feed conversion ratio (g feed/g gain)	14–35 days	1.48	1.53	1.48	0.044	0.9290
	35–42 days	1.77	1.83	1.78	0.035	0.7981
	14–42 days	1.55	1.58	1.55	0.034	0.8685

^a, b, c^ Means in the same line with different superscripts differ significantly (*p* < 0.05). SEM = standard error of the means.

**Table 3 animals-10-00958-t003:** Effect of dietary SAB powder on serum biochemical parameters (average values/group).

Parameter	SAB 0%	SAB 0.025%	SAB 0.05%	SEM	*p*-Value
*Energy profile*					
Glucose, mg/dL	245.04 ^a^	219.06 ^b^	195.01 ^b^	9.272	0.0591 *
Cholesterol, mg/dL	97.41	86.16	93.19	3.261	0.5397
Triglycerides, mg/dL	38.82	38.57	36.57	1.418	0.668
*Hepatic paramters*					
Total protein (g/dL)	2.40	2.28	2.15	0.066	0.2762
Total bilirubin (mg/dL)	0.06	0.06	0.05	0.002	0.1616
ALT ^1^ (TGP), U/L	5.06 ^a^	4.34 ^a^	2.98 ^b^	2.66	0.0147
AST ^2^ (TGO), U/L	375.6 ^a^	361.5 ^a^	248.4 ^b^	25.57	0.0350

^a, b^ Means in the same line with different superscripts differ significantly (*p* < 0.05). SEM = standard error of the means; ^1^ Alanine aminotransferase; ^2^ aspartate aminotransferase; *n* = 10. * ANOVA value of *p* was not significant, the superscripts are declared according to the Fischer test results.

**Table 4 animals-10-00958-t004:** Effect of dietary SAB powder on oxidative stress biomarkers in liver tissue.

Parameter	SAB 0%	SAB 0.025%	SAB 0.05%	SEM	*p*-Value
TBARS ^1^ (nmol g/tissue)	48.1 ^a^	35.4 ^b^	35.6 ^b^	1.639	˂0.0001
PC ^2^ (nmol g/tissue)	27.34 ^a^	23.5 ^b^	21.17 ^b^	0.806	0.0014
TAC ^3^ (% Free radical scavenging)	49.38 ^a^	50.78 ^a^	64.41 ^b^	2.125	0.0011
GSH ^4^ (μmol g/tissue)	2.59 ^a^	2.67 ^a^	3.38 ^b^	0.113	0.0013
SOD ^5^ (U g/tissue)	1898.75	1596.58	1719.08	66.301	0.1772

^a, b^ Means in the same line with different superscripts differ significantly (*p* < 0.05). SEM = standard error of the means; *n* = 10; ^1^ Thiobarbituric reactive substances; ^2^ Protein oxidation; ^3^ Total antioxidant capacity; ^4^ Glutathione; ^5^ Superoxide dismutase.

**Table 5 animals-10-00958-t005:** Effects of dietary SAB powder on intestinal and caecal bacterial populations (lg10 CFU/g wet intestinal/caecal content).

Parameter	SAB 0%	SAB 0.025%	SAB 0.05%	SEM	*p*-Value
*Intestinal content*					
*E. coli*	6.361 ^a^	6.359 ^a^	6.357 ^b^	0.001	0.0100
Staphylococci	6.169 ^a^	6.165 ^ab^	6.163 ^b^	0.001	0.0118
Lactobacilli	7.404 ^a^	7.412 ^b^	7.419 ^c^	0.002	˂0.0001
*Salmonella spp.*	absent	absent	absent	NA	NA
*Caecal content*					
*E. coli*	10.191 ^a^	10.190 ^a^	10.174 ^b^	0.003	<0.0002
Staphylococci	8.836 ^a^	8.829 ^a^	8.819 ^b^	0.003	0.0013
Lactobacilli	11.564 ^a^	11.637 ^b^	11.665 ^c^	0.011	<0.0001
*Salmonella spp.*	absent	absent	absent	NA	NA

^a, b, c^ Means in the same line with different superscripts differ significantly (*p* < 0.05). NA: not calculated, *n* = 10.

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
