# Peer review of "Influence of Dietary Supplementation of Salix alba Bark on Performance, Oxidative Stress Parameters in Liver and Gut Microflora of Broilers"

_animals, 2020, doi:10.3390/ani10060958_

Round 1

Reviewer 1 Report

This study provided interesting results regarding the plant polyphenols from willow bark on growth and antioxidant capacity in broiler chicks. The study is well design and prepared. Publication is recommended after minor revision.

  1. Abstract:“cellular injury caused by oxidative stress” in the conclusion sentence, please rephrase. Because there was no challenge and broilers were raised under normal condition.
  2. Table 1, please check the CP level for 0.05% SAB group at grower phase, 19.92 is far away from 22.41 and 22.19.What is the reason for the difference?  Were the values calculated or analyzed ?
  3. Please list the lysine, methionine plus cystine, threonine, and tryptophan values in Table 1. For lysine in the ingredient, was it in hydrochloride or sulphate form?
  4. Statistical analysis, please make it clear themultiple comparison method you used to separate means.
  5. For ADFI during 35-42 d of age, there were differences among groups. Thus, it is not appropriate to say “no difference of growth performance” in the second paragraph of the Discussion section.
  6. For the reduction of glucose, you can discuss form the glucose homeostasisor diabetic modulation effects of polyphenol in medical study.
  7. The last line at page 8, aminoacids should be amino acids.
  8. Line 3 of page 9, her oxidation, please rephrase.
  9. GSH enzyme activity, please rephrase. GSH is peptide not enzyme.
  10. Please include dosage discussion, within your study and between other studies. What is the reason for better growth and antioxidant exhibition in500 ppm group than 250 ppm group. Grape polyphenols or green tea polyphenol may take action at much lower dosage, it is better to include comparison across studies. 

Author Response

Response to Reviewer 1 Comments

Observation 1. Abstract:“cellular injury caused by oxidative stress” in the conclusion sentence, please rephrase. Because there was no challenge and broilers were raised under normal condition.

Answer: We agree the observation made and we modified accordingly (page 1, simple summary and in abstract conclusion)

Observation 2. Table 1, please check the CP level for 0.05% SAB group at grower phase, 19.92 is far away from 22.41 and 22.19.What is the reason for the difference? Were the values calculated or analyzed ?

Answer: As reviewer suggests we checked the table 1 content. Unfortunately, we noticed a major mistake produced and we replaced the analysis table content accordingly. We performed quality control determinations of feeds but the table 1 contains calculated data. The diet formulations were calculated using HYBRIMIN Futter 5 Nutrition programme and were used the energy and crude protein as fixed factors, according to nutritional requirements of broilers published by NRC (Table 1).

Observation 3: Please list the lysine, methionine plus cystine, threonine, and tryptophan values in Table 1. For lysine in the ingredient, was it in hydrochloride or sulphate form?

Answer: We agree the observation made and we modified accordingly (Table 1).

Observation 4: Statistical analysis, please make it clear the multiple comparison method you used to separate means.

Answer: We agree the observation made and we modified accordingly (page 5, cap.2.8. Statistical analysis).

Observation 5: For ADFI during 35-42 d of age, there were differences among groups. Thus, it is not appropriate to say “no difference of growth performance” in the second paragraph of the Discussion section.

Answer: We agree the observation made and we modified accordingly (page 8, cap. 4 Discussion, paragraph 2)

Observation 6: For the reduction of glucose, you can discuss form the glucose homeostasisor diabetic modulation effects of polyphenol in medical study.

Answer: We agree the observation made and we modified accordingly (page 9, cap. 4 Discussion, paragraph 1)

Observation 7: The last line at page 8, aminoacids should be amino acids.

Answer: We agree the observation made and we modified accordingly (page 9, paragraph 3).

Observation 8: Line 3 of page 9, her oxidation, please rephrase.

Answer: We agree the observation made and we modified accordingly, replacing ,,her’’ with ,,its’’ (page 9, paragraph 3).

Observation 9: GSH enzyme activity, please rephrase. GSH is peptide not enzyme.

Answer: We agree the observation made and we rephrased accordingly (page 9-10, paragraph 4).

Observation 10: Please include dosage discussion, within your study and between other studies. What is the reason for better growth and antioxidant exhibition in500 ppm group than 250 ppm group. Grape polyphenols or green tea polyphenol may take action at much lower dosage, it is better to include comparison across studies. 

Answer: We agree the observation made and we rephrased accordingly (page 8, paragraph 2).

Reviewer 2 Report

Page 1, abstract "The performance parameters did not register significant differences between treatments" this is not completely true: Average daily feed intake is statistically different

Table 1 - Crude protein, % of group SAB 0.05% (19,92) is wrong, please correct

Table 2 - Average daily feed intake 14-35 days , p-value 0.0335, different superscripts letters are missing

Table 2 - Average daily feed intake  14-42 days, p-value 0.0301, different superscripts letters are missing

Table 2 - Average daily weight gain 14-35 days, p-value 0.6965 and average values with different superscripts

Table 2 - Average daily weight gain 14-42 days, p-value 0.3444 and average values with different superscripts

Tables 2, 3, 4, 5- change "a, b,c Means in the same column with different superscripts" as "a, b,c Means in the same line with different superscripts"

Page 6 (cap. 3.2)- Only in the last week of the trial (35-42 days) the ADFI was significantly higher (p<0.05) in SAB 0.05% treatment compared  with both SAB 0% and SAB 0.025%. Throughout the entire experimental trial, no significant differences were recorded between treatments (Table 2). No significant differences were noticed for SAB 0.025% and SAB 0.05% compared to SAB 0% treatment regarding ADWG, through entire experimental period (Table 2). Change according to the values in table 2.

Page 6 (cap. 3.2)  - Even though in the last week of the trial the broilers from SAB 0.05% group had a higher ADFI than the other groups, the FCR was similar to SAB 0% and lower than SAB 0.025% (Table 2). Change according to the values in table 2.

Page 7 (cap. 3.4) " The dietary inclusion of the SAB powder inhibited protein oxidation, but the lowest concentration of PC was found in liver tissue of the chicken fed SAB 0.05% treatment. " change, also SAB 0.025 is significantly different from group SAB 0.

Page 8, discussion " Regarding the performance results, there were no difference between experimental treatments (SAB 0.025%, SAB 0.05%) and SAB 0% (Table 2). Delete or modify: in Table 2 there are several differences. It would be appropriate to explain why the SAB 0.025% group showed lower perfromance (sometimes also significant) in comparison to SAB 0 group.

Page 9 after The ingestion of natural antioxidants has been associated with an increase in the cell’s antioxidant activity. please add the reference (Gheisar & Kim, 2018)

References

Gheisar,M.M. and Kim,I.H. Phytobiotics in poultry and swine nutrition. Italian Journal of Animal Science. 201, 17:1, 92-99.

Author Response

Response to Reviewer 2 Comments

Observation 1. Page 1, abstract "The performance parameters did not register significant differences between treatments" this is not completely true: Average daily feed intake is statistically different.

Answer: We agree the observation made and we modified accordingly (page 1, abstract section)

Observation 2. Table 1 - Crude protein, % of group SAB 0.05% (19,92) is wrong, please correct.

Answer: We agree the observation made. As reviewer suggests we checked the table 1 content. Unfortunately, we noticed a major mistake produced and we replaced the analysis table content accordingly. We performed quality control determinations of feeds but the table 1 contains calculated data. The diet formulations were calculated using HYBRIMIN Futter 5 Nutrition programme and were used the energy and crude protein as fixed factors, according to nutritional requirements of broilers published by NRC.

Observation 3: Table 2 - Average daily feed intake 14-35 days, p-value 0.0335, different superscripts letters are missing

Answer: We agree the observation made, but by mistake some p- values were inversed. The superscripts expressing the statistically significance are correct inserted. Only the p-values were wrong. Consequently, we modified accordingly (Table 2)

Observation 4: Table 2 - Average daily feed intake 14-42 days, p-value 0.0301, different superscripts letters are missing.

Answer: We agree the observation made, but by mistake some p- values were inversed. The letters expressing the statistically significance are correct. Only the p-values were wrong. Consequently, we modified accordingly (Table 2)

Observation 5: Table 2 - Average daily weight gain 14-35 days, p-value 0.6965 and average values with different superscripts.

Answer: We agree the observation made, but by mistake some p- values were inversed. The letters expressing the statistically significance are correct. Only the p-values were wrong. Consequently, we modified accordingly (Table 2)

Observation 6: Table 2 - Average daily weight gain 14-42 days, p-value 0.3444 and average values with different superscripts

Answer: We agree the observation made, but by mistake some p- values were inversed. The letters expressing the statistically significance are correct. Only the p-values were wrong. Consequently, we modified accordingly (Table 2)

Observation 7: Tables 2, 3, 4, 5- change "a, b,c Means in the same column with different superscripts" as "a, b,c Means in the same line with different superscripts"

Answer: We agree the observation made and we modified accordingly (Tables 2, 3, 4, 5).

Observation 8: Page 6 (cap. 3.2)- Only in the last week of the trial (35-42 days) the ADFI was significantly higher (p<0.05) in SAB 0.05% treatment compared with both SAB 0% and SAB 0.025%. Throughout the entire experimental trial, no significant differences were recorded between treatments (Table 2). No significant differences were noticed for SAB 0.025% and SAB 0.05% compared to SAB 0% treatment regarding ADWG, through entire experimental period (Table 2). Change according to the values in table 2.

Answer: We agree the observation made, but p-values were inversed, thus superscripts are correct inserted. Consequently, the ideas presented in the paragraph are correct (page 6, paragraph 2, cap. 3.2).

Observation 9: Page 6 (cap. 3.2) - Even though in the last week of the trial the broilers from SAB 0.05% group had a higher ADFI than the other groups, the FCR was similar to SAB 0% and lower than SAB 0.025% (Table 2). Change according to the values in table 2.

Answer: We agree the observation made, but p-values were inversed, thus superscripts are correct inserted. Consequently, the ideas presented in the paragraph are correct (page 6, paragraph 2, cap. 3.2). In order to be clearer, we mention the p-value in the text (page 6, paragraph 2, cap. 3.2).

Observation 10: Page 7 (cap. 3.4) " The dietary inclusion of the SAB powder inhibited protein oxidation, but the lowest concentration of PC was found in liver tissue of the chicken fed SAB 0.05% treatment. " change, also SAB 0.025 is significantly different from group SAB 0.

Answer: We agree the observation made and we rephrased accordingly (page 7, cap 3.4).

Observation 11: Page 8, discussion " Regarding the performance results, there were no difference between experimental treatments (SAB 0.025%, SAB 0.05%) and SAB 0% (Table 2). Delete or modify: in Table 2 there are several differences. It would be appropriate to explain why the SAB 0.025% group showed lower perfromance (sometimes also significant) in comparison to SAB 0 group.

Answer: We agree the observation made and we rephrased accordingly (page 8, discussion). Regarding the explanation for a lower performance of SAB 0.025% group in comparison to SAB 0% group, is difficult to be stated, because all three groups were raised under the same conditions, were fed the same diet during starter stage (1-14 days). For this reason, further, we want to conduct a study on investigating the negative potential of this dose of SAB on broiler performance and the mechanism responsible for this effect.

Observation 12: Page 9 after The ingestion of natural antioxidants has been associated with an increase in the cell’s antioxidant activity. please add the reference (Gheisar & Kim, 2018)

Answer: We agree the suggestion made and we added the reference (page 9, discussion, reference 44.).

Reviewer 3 Report

- The manuscript needs a revision by a native English speaker. Some specific points for improvement are given below.

- Page 2, Paragraph 1. Please consider changing text “it is needed to search for solution” to text “research is needed for solutions”.

- Page 2, Materials and methods. Paragraph 1. Please write the name of the institute that gave the certification. Also, add the EU Directive as reference

- Page 2, Chapter 2.1. Paragraph 1. Please consider changing text “took from” to “that were provided by”.

- Page 2, Chapter 2.1. Paragraph 1. Please consider changing text “it was used wood shaving” to “wood shaving were used”.

- Page 2, Chapter 2.1. Paragraph 1. Please change format of “NRC (1994)” reference to numerical i.e. reference number [13]

- Page 2, Chapter 2.1. Paragraph 1. Authors describe the feed supplement as Salix Alba bark powder in most of the text. This is also the name of the supplement in the title of the manuscript. However, the supplement used is an hydro-alcoholic extract of the plant containing nearly pure (98%) salicin. In my opinion such as product cannot be considered as a plant “bark powder”, but a nearly pure chemical substance i.e. salicin. Of course, this means that the title and the whole manuscript should be reformatted accordingly.

- Page 4, Chapter 2.3. Which part of the intestine was used for the content collection apart from the caecum?

- Page 5, Chapter 2.5. Please add additional information for the analyser: Company, Country.

- Page 5, Chapter 2.6.5. Please add additional information for the kit: Company, Country.

- Pages 5-6, Table 2: Check the statistical analysis in this table. In the average daily feed intake analysis for days 14-35 & 14-42 the P values are significant (<0.05) but no superscript are shown for the differences between the treatments. Moreover, on the average daily weight gain analysis for days 14-35 & 14-42 superscripts are given for values that do not differ. Also, revise the text in the results section, accordingly. Also, in the footnote it should be written that values in the same row (not column) with different superscripts differ and this correction should be made in all other tables.

Furthermore, the fact that the 0.025% supplementation lowered the body weight and daily gain, whereas the 0.050% increased these parameters is unexpected. How do the authors explain these findings?

- Page 6, Table 3: Please notice that the glucose P value no significant (P>0.05) therefore there should be no differences between the treatments.

- Discussion and conclusions: Since the product is 98% salicin, perhaps it would be better to focus on this substance, instead of talking about phenolic content in general in order to explain and discuss the results of the trial.

- References section. A revision of this section is suggested, because the format of the references is not uniform. For some examples, check the use of “and” before last author (Ref. 4); the capitalization of article titles (Ref. 2); The format of presenting DOI (with or without HTTP; Ref 12).

Author Response

Response to Reviewer 3 Comments

Observation 1. - Page 2, Paragraph 1. Please consider changing text “it is needed to search for solution” to text “research is needed for solutions”.

Answer: We agree the observation made and we modified accordingly (page 2, introduction, paragraph 1)

Observation 2. - Page 2, Materials and methods. Paragraph 1. Please write the name of the institute that gave the certification. Also, add the EU Directive as reference.

Answer: We agree the observation made, and we added the reference (page 2, materials and methods, reference no. 13) and the name of the institute.

Observation 3: - Page 2, Chapter 2.1. Paragraph 1. Please consider changing text “took from” to “that were provided by”.

Answer: We agree the observation made and we modified accordingly (page 2, chapter 2.1, paragraph 1).

Observation 4: Page 2, Chapter 2.1. Paragraph 1. Please consider changing text “it was used wood shaving” to “wood shaving were used”.

Answer: We agree the observation made and we modified accordingly (page 2, chapter 2.1, paragraph 1).

Observation 5: Page 2, Chapter 2.1. Paragraph 1. Please change format of “NRC (1994)” reference to numerical i.e. reference number [13]

Answer: We agree the observation made and we modified accordingly (page 2, chapter 2.1, paragraph 1, reference no. 14).

Observation 6: - Page 2, Chapter 2.1. Paragraph 1. Authors describe the feed supplement as Salix Alba bark powder in most of the text. This is also the name of the supplement in the title of the manuscript. However, the supplement used is an hydro-alcoholic extract of the plant containing nearly pure (98%) salicin. In my opinion such as product cannot be considered as a plant “bark powder”, but a nearly pure chemical substance i.e. salicin. Of course, this means that the title and the whole manuscript should be reformatted accordingly.

Answer: White willow bark extract is the commercial name of the product. The extract purchased and included in the diets of animals was a white powder. We verify the producer details about product and we found some mistakes in our report. The producer mentioned a solvent extraction for powder obtaining and we considered as wrong the hydro-alcoholic extract formulation. We rephrased the sentences and removed the hydro-alcoholic extract formulation. Also, the salicin content of powder is 25% - 98% according to producer’s details. Taking into account that we cannot determined the salicin content but the antioxidant capacity and total polyphenols were determined (data published by Saracila et al., 2019), we considered that some effects observed in our results can be attributed to other bioactive compounds of Salix Alba extract.

Saracila, M., et al. "Use of a hydroalcoholic extract of Salix alba L. bark powder in diets of broilers exposed to high heat stress." South African Journal of Animal Science 49.5 (2019): 942-954.

Observation 7: - Page 4, Chapter 2.3. Which part of the intestine was used for the content collection apart from the caecum?

Answer: We agree the observation made and we inserted in the text (page 4, chapter 2.3).

Observation 8: - Page 5, Chapter 2.5. Please add additional information for the analyser: Company, Country.

Answer: We agree the observation made and we added the additional information (page 4, chapter 2.5).

Observation 9: - Page 5, Chapter 2.6.5. Please add additional information for the kit: Company, Country.

Answer: We agree the observation made and we added the additional information (page 5, chapter 2.6. 5).

Observation 10: - Pages 5-6, Table 2: Check the statistical analysis in this table. In the average daily feed intake analysis for days 14-35 & 14-42 the P values are significant (<0.05) but no superscript are shown for the differences between the treatments. Moreover, on the average daily weight gain analysis for days 14-35 & 14-42 superscripts are given for values that do not differ. Also, revise the text in the results section, accordingly. Also, in the footnote it should be written that values in the same row (not column) with different superscripts differ and this correction should be made in all other tables.

Answer: We agree the observation made but by mistake some p- values were inversed. We verified, and the superscripts expressing the statistically significance are correct. Only the p-values were wrong. Consequently, we modified accordingly both in the table (Table 2) and in the text. We modified also in the footnote the statistical expression.

Observation 11: Furthermore, the fact that the 0.025% supplementation lowered the body weight and daily gain, whereas the 0.050% increased these parameters is unexpected. How do the authors explain these findings?

Answer: We agree the observation made. We found that the product studied as supplement in broiler diet possess a negative effect on broiler performance at a lower dose of inclusion. The explanation is difficult to be stated, because all three groups were raised under the same conditions, were fed the same diet during starter stage (1-14 days). For this reason, further, we want to conduct a study on investigating the negative potential of this dose of SAB on broiler performance and the mechanism responsible for this effect.

Observation 12: - Page 6, Table 3: Please notice that the glucose P value no significant (P>0.05) therefore there should be no differences between the treatments.

Answer: In the table is declared the p value resulted from Anova test. The superscripts are declared according to the Fischer test results.

Observation 13: Discussion and conclusions: Since the product is 98% salicin, perhaps it would be better to focus on this substance, instead of talking about phenolic content in general in order to explain and discuss the results of the trial.

Answer: Same answer as observation 6.

Observation 14: References section. A revision of this section is suggested, because the format of the references is not uniform. For some examples, check the use of “and” before last author (Ref. 4); the capitalization of article titles (Ref. 2); The format of presenting DOI (with or without HTTP; Ref 12).

Answer: We agree the observation made and we revised the references format. Because in the instructions for authors the format of reference is without ,,and’’ before last author, we modified accordingly. Also, we modified the format of DOI, using the form without ,,http’’ and we used the lowercase for the article titles.

Round 2

Reviewer 3 Report

- Regarding the response of the authors about the salicin content, I would like to comment that it is unexpected for a standardized product to have such large variability of salicin, ranging from 25% to 98%. Perhaps these values should be rechecked.

- Page 2, last paragraph. Please change text “25- 98%” to text “25% - 98%”.

- Table 3, footnote. It is suggested to clarify in the footnote that although the ANOVA value of P is not significant for the glucose analysis (P=0.0591), the post-hoc test showed significant differences between the groups.

Author Response

Dear Editor,

We would like to thank the referee for the close reading and for the proper suggestions. We hope that we provided all the answers to the reviewers’ comments.

Thank you very much for taking into account to publish our manuscript. The present version of the paper has been revised according to the reviewers and editor suggestions. All the changes were marked with yellow.

We uploaded the corrected version of the article according to reviewers and editor suggestions. Also, we attach the response to reviewer.

                We look forward to hear from you soon.

Sincerely yours,

Tatiana Dumitra PANAITE et al.

Regarding the response of the authors about the salicin content, I would like to comment that it is unexpected for a standardized product to have such large variability of salicin, ranging from 25% to 98%. Perhaps these values should be rechecked.

Response: Thank you for your comment. We consider your suggestion and we try to find a way to determine the salicin content of the product. Unfortunately, the large variability of the salicin content is declared by the producer. Also, in the web sources we found a large variability of salicin (15%, 15.2%, 25%, 53-65%) for many standardized products.

Page 2, last paragraph. Please change text “25- 98%” to text “25% - 98%”.

Response: According to reviewer’s suggestion, we replaced “25- 98%” with “25% - 98%”.

Table 3, footnote. It is suggested to clarify in the footnote that although the ANOVA value of P is not significant for the glucose analysis (P=0.0591), the post-hoc test showed significant differences between the groups.

Response: The table 3 footnote was modified with: ” ANOVA value of P was not significant, the superscripts are declared according to the Fischer test results”